# Evaluation of Extreme Cold and Drought over the Mongolian Plateau

**Zhaofei Liu [1],[*] , Zhijun Yao [1], Heqing Huang [1] , Batbuyan Batjav [2] and Rui Wang [1],***

1    Institute of Geographic Sciences and Natural Resources Research, Chinese Academy of Sciences,
     Beijing 100101, China; yaozj@igsnrr.ac.cn (Z.Y.); huanghq@igsnrr.ac.cn (H.H.)
2    Institute of Geography, Mongolian Academy of Sciences, Ulaanbaatar 210620, Mongolia;
     b_batbuyan@yahoo.com
*    Correspondence: zfliu@igsnrr.ac.cn (Z.L.); wangr@igsnrr.ac.cn (R.W.); Tel.: +86-10-6488-9527 (Z.L.)

**Abstract:** Extreme cold and meteorological drought in the Mongolian Plateau (MP) were investigated during 1969–2017. Several drought indices were evaluated by analyzing recorded historical drought data in the Chinese region of the MP. The evaluated drought indices were then applied to detect drought characteristics in the entire MP. The trends of extreme cold indices showed that the climate of the MP has warmed during the past 49 years; however, the frequency of cold day/night has increased in the Mongolian region. The climate of Mongolia has also become colder in the spring season. The comprehensive meteorological drought index (CMDI) and the standardized precipitation index with a six-month scale (SPI6) exhibited better performances, showing high consistency between the spatial patterns of the two indices. However, drought represented by the SPI6 was enhanced greater than that expressed by the CMDI. Drought in the MP has been enhanced during the past 49 years, particularly in the Ordos and Alashan plateaus and the Xiliao River basin in China. Moreover, drought has been enhanced from August to October, particularly in the Mongolian region. However, spring drought has shown a weakening trend, which has been beneficial for agriculture and husbandry sectors in some regions of the MP.

**Keywords:** extreme climate; meteorological drought; comprehensive meteorological drought index (CMDI); standardized precipitation index (SPI); empirical orthogonal function (EOF); Mann–Kendall test

## 1. Introduction

Climate change is one of the great environmental issues of the 21st century. However, variations and trends in extreme climate are more sensitive to climate change than the mean values and thus have received significantly more attention [1]. Extreme weather and climate events have received increased attention because they are associated with high losses of human life and increasing expenses [2,3]. The greatest threat to humans and the natural environment is manifested locally via changes in regional extreme weather and climate events [4,5]. Changes in the frequency or intensity of extreme weather and climate events have profound impacts on both human society and the natural environment [1] because society as a whole is vulnerable to extreme weather and climate [6].

The Mongolian Plateau (MP) is located in an arid to semi-arid region. Its annual precipitation is only 246.1 mm. In contrast, annual potential evaporation (PE) is 986.4 mm, which significantly increases the aridity index. The MP has experienced several extreme climate events during the past decades, including severe extreme cold and droughts [7,8]. The consecutive 1999–2002 droughts and dzuds were the worst recorded during the last 50 years and caused 30% of the national herd losses in Mongolia [9]. The 2009–2010 dzud was also very severe in which 8.5 million livestock died in Mongolia, amounting to 20% of the national herd [10]. The frequency and magnitude of these extreme events

have increased during the 2000–2010 period, compared with those recorded in a few decades prior to 2000 [11], and are expected to increase with future climate changes [10]. Therefore, it is meaningful to investigate extreme cold and drought over the MP.

Various approaches have been taken to address the potential changes in extreme climate. The World Meteorological Organization Commission for Climatology/Climate Variability held a meeting in Geneva in November 1999, and was the first to recommended 10 simple and feasible indices for climate extremes [12]. A number of indices of climate extremes have been developed for easy calculation and application in different parts of the world [13]. It has been suggested that the worldwide use of accepted climate extreme indices should allow for comparisons with associated information from various regional-scale studies and provide evidence of changes in extreme weather and climate events [14]. The most frequently used of these indices is the Expert Team on Climate Change Detection and Indices (ETCCDI), which have been recommended by the Royal Netherlands Meteorological Institute. In this study, ETCCDI indices are used to evaluate the extreme cold of the MP by percentile and frequency of daily air temperature.

There are many drought indices that have been proposed [15]. These indices can be categorized into three forms, the meteorological, hydrological, and agricultural drought indices. In this study, we focus on meteorological drought. Some popular meteorological drought indices, which are calculated by climate data, include the standardized precipitation index (SPI), Palmer drought severity index (PDSI), standardized precipitation evapotranspiration index, and the regional drought area index. Each index has particular strengths and weaknesses [16]. The SPI and the PDSI are more popular than other indices [17,18]. Keyantash and Dracup [15] evaluated these meteorological drought indices by applying a weighted set of six evaluation criteria and found that the SPI showed better performance than the PDSI. Hayes et al. [17] found that the SPI improved drought detection and monitoring capabilities over the PDSI when monitoring the 1996 drought in the United States. The SPI is an ideal candidate for drought risk analysis due to its intrinsic probabilistic nature [19]. It has several advantages in terms of great flexibility, less complexity, and statistical consistency [20]. The length of precipitation records for SPI calculation is a continuous period of at least 30 years [21]. Because the SPI is a probability-related index, longer record lengths relate to greater confidence in the stability of the underlying statistics. Therefore, it is recommended that the length of precipitation data used in SPI calculation should be as long as possible [22].

In this study, extreme cold and meteorological drought in the MP are evaluated by meteorological observed data and statistical recorded historical data. Firstly, several extreme temperature indices, defined by ETCCDI, are calculated as extreme cold indices. Trends of extreme cold indices are detected by using a non-parametric Mann–Kendall test method. Their spatial distribution is also investigated. Secondly, drought indices, including multiple timescales of the SPI and the comprehensive meteorological drought index (CMDI), are calculated and evaluated by using recorded historical drought data in the Chinese region of the MP. Finally, the evaluated drought indices that performed better in the Chinese region were applied for detecting drought characteristics of the entire MP. Drought characteristics include spatial patterns, temporal characteristics, and trends.

## 2. Materials and Methods

### 2.1. Study Area Description

The MP includes the entire country of Mongolia in addition to the entire Inner Mongolia Autonomous Region and parts of Gansu, Ningxia, and Shaanxi provinces in China (Figure 1). It is located at an arid to semi-arid region and has a continental climate. The monthly mean air temperature ranges from $-15.2\ ^{\circ}\text{C}$ in January to $21.6\ ^{\circ}\text{C}$ in July. It is lower than $0\ ^{\circ}\text{C}$ from November to March of the following year. The monthly maximum air temperature is even lower than $0\ ^{\circ}\text{C}$ in the winter season (DJF—December, January and February). As previously mentioned, the precipitation rate is

significantly lower than the PE in the MP. Precipitation occurs mainly from May to October, which takes up more than 90% of the annual precipitation.

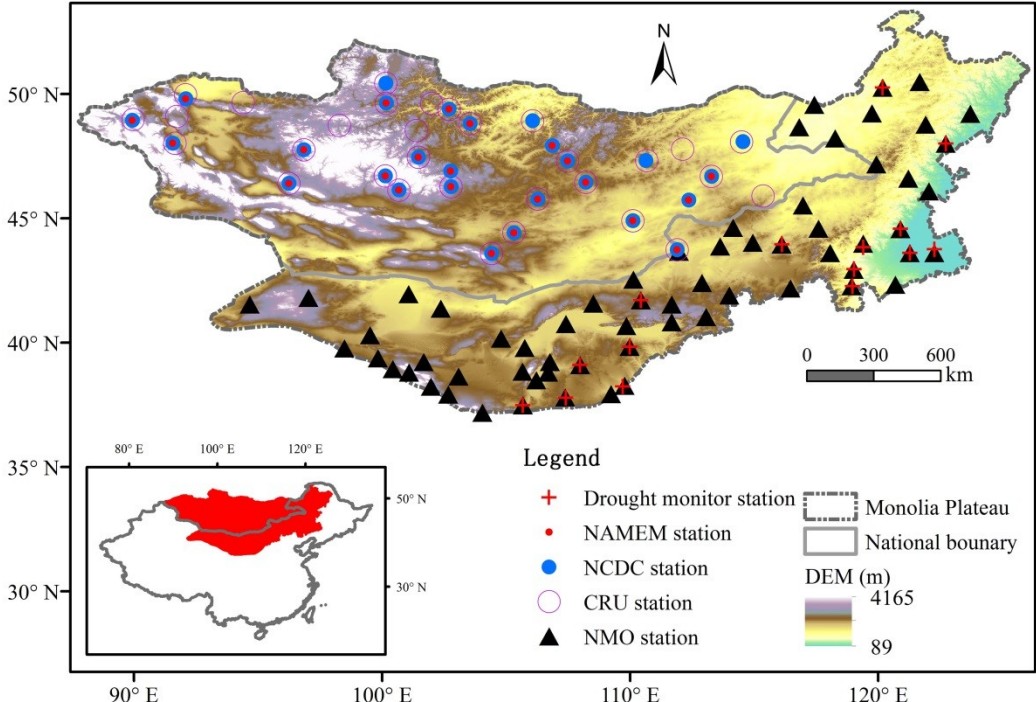

**Figure 1.** Location of the Mongolian Plateau, meteorological stations, and drought monitor stations (NAMEM: National Agency for Meteorology and Environment Monitoring; NCDC: National Climatic Data Center; CRU: Climatic Research Unit; NMO: National Meteorological Observatory).

*2.2. Data Description*

2.2.1. Meteorological Observed Station Data

Observed daily maximum air temperature (TX) and minimum air temperature (TN) were used for extreme cold analyses, and observed monthly precipitation was applied for drought analyses. In the Chinese region of the MP, 64 National Meteorological Observatory (NMO) stations provided by the National Climate Center, China Meteorological Administration (CMA), include continuous data series from 1961 to 2017. All observed data series have passed NMO data quality control. In the Mongolian region of the MP, 27 air temperature observation stations are operated by the National Oceanic and Atmospheric Administration (NOAA) National Climatic Data Center (NCDC; ftp://ftp.ncdc.noaa.gov/pub/data/gsod), which provides continuous daily data series from 1969 to 2017. In addition, 31 precipitation stations are operated in the Mongolian region of the MP by the Climatic Research Unit (CRU; https://crudata.uea.ac.uk/), which provides continuous monthly data series from 1964 to 2017. Observed precipitation data series at 23 stations were obtained from the National Agency for Meteorology and Environment Monitoring (NAMEM), Mongolia. It only includes annual precipitation data series for the period 1980–2003. We evaluated the CRU and the NCDC datasets by the NMO and the NAMEM dataset. Results show that the CRU and the NCDC data are consistent with the NMO and the NAMEM data. Based on the data evaluation, we believe that the NCDC and the CRU data have reliability at the study region. The geographical locations and spatial distribution of these stations are shown in Figure 1.

2.2.2. Recorded Historical Drought Data

Two recorded historical drought datasets were used in this study. The first includes the area affected by drought in the Inner Mongolia Autonomous Region, which was obtained from the National

Bureau of Statistics of China. It includes statistics of the drought affected area for each province (municipality). The total annual data series record was from 1979 to 2017. Annual data series of 1979–1995 were obtained from the 2010 Statistical Yearbook of the Republic of China, whereas those of 1996–2017 were obtained from the website http://www.stats.gov.cn/tjsj/ndsj/. The second includes recorded drought disaster event data extracted from the agrometeorological disaster dataset of China, which were provided by the CMA. In this study, 15 drought monitoring stations were used. The spatial distribution of these stations is shown in Figure 1. These stations are relatively well distributed in the Inner Mongolia Autonomous Region. The observation network covers most of the area of the region. The recorded data of these stations includes affected crops, occurrence time, severity, and affected area for each drought event. However, because drought severity and area in these datasets have qualitative descriptions, only the number of drought events was applied as this second drought dataset.

*2.3. Methodology Description*

2.3.1. Extreme Cold Indices

In this study, six extreme temperature indices (Table 1) recommended by the ETCCDI, available at http://etccdi.pacificclimate.org/list_27_indices.shtml, were applied to investigate extreme cold in the MP. The indices were chosen for assessing changes in intensity and frequency of cold events and included three different categories: (1) absolute indices represented maximum or minimum values within a month (TXn or TNn); (2) percentile-based indices, such as occurrences of cold days or nights (TX10p or TN10p); and (3) threshold indices, defined as the number of days in which a temperature value was above or below a fixed threshold of frost day (FD) or ice day (ID). In addition, annual mean value of daily maximum or minimum temperature (TX_Ann or TN_Ann) was used.

**Table 1.** Definitions of six extreme cold indices used in this study.

| Index | Descriptive Name | Definition | Unit |
|-------|-----------------|------------|------|
| TXn | Coldest day | Minimum value of daily maximum temperature | °C |
| TNn | Coldest night | Minimum value of daily minimum temperature | °C |
| TX10p | Cold day frequency | Percentage of days when TX < 10th percentile | % |
| TN10p | Cold night frequency | Percentage of days when TN < 10th percentile | % |
| FD | Frost days | Annual count of days when TN (daily minimum temperature) < 0 °C | days |
| ID | Icing days | Annual count of days when TX (daily maximum temperature) < 0 °C | days |

2.3.2. Drought Indices

The SPI was applied for drought detection because it quantifies the precipitation deficit for multiple timescales. Different timescales of the index express various types of droughts. For example, the SPI with shorter timescales represents agricultural and meteorological droughts, whereas that with longer timescales expresses hydrological drought. In this study, the SPI was calculated on 1-, 3-, 6-, and 12-month time scales, which correspond respectively to the past 1, 3, 6, and 12 months of observed total precipitation. These multiple SPI timescales were expressed as SPI1, SPI3, SPI6, and SPI12, respectively. Table 2 shows the drought classification based on the SPI index. The SPI program was obtained from the National Drought Mitigation Center (http://drought.unl.edu).

**Table 2.** Classification of drought by the SPI and CMDI indices.

| Drought Classification | SPI | CMDI |
|------------------------|-----|------|
| Non-drought | $-0.5 <$ SPI | $-0.6 <$ CMDI |
| Mild drought | $-1.0 <$ SPI $\leq -0.5$ | $-1.2 <$ CMDI $\leq -0.6$ |
| Moderate drought | $-1.5 <$ SPI $\leq -1.0$ | $-1.8 <$ CMDI $\leq -1.2$ |
| Severe drought | $-2.0 <$ SPI $\leq -1.5$ | $-2.4 <$ CMDI $\leq -1.8$ |
| Extreme drought | SPI $\leq -2.0$ | CMDI $\leq -2.4$ |

The CMDI used in this study is recommended by the National Standard of China (GB/T 20481-2006) for classification of meteorological drought [23]. This index, also referred to as the composite index [24], is used for drought detection in China by the Public Meteorological Service Center of CMA. It was synthesized by the SPI and the relative moisture index by

$$CMDI = a \times SPI1 + b \times SPI3 + c \times M \tag{1}$$

$$M = (P - PE)/PE \tag{2}$$

where M is the relative moisture index, and P and PE are precipitation and potential evaporation, respectively. PE was calculated by using the Penman–Monteith method, which is recommended by the United Nations Food and Agriculture Organization (FAO) [25]. Four CRU precipitation stations had no other meteorological data. The PE at these stations was interpolated from that of adjacent NCDC stations by using the inverse distance weighted method. Constants a, b, and c in Equation (1) were taken as 0.4, 0.4, and 0.8, respectively [23].

The Penman–Monteith method for estimating $ET_0$ can be derived as follows:

$$ET_0 = \frac{0.408\Delta(R_n - G) + \gamma \frac{900}{T+273} u_2 (e_s - e_a)}{\Delta + \gamma(1 + 0.34u_2)} \tag{3}$$

where,

$ET_0$—reference crop evapotranspiration [mm/day],
$R_n$—net radiation at the crop surface [MJ/m$^2$ day],
$G$—soil heat flux density [MJ/m$^2$ day],
$T$—mean daily air temperature at 2-m height [°C],
$u_2$—wind speed at 2-m height [m/s],
$e_s$—saturation vapor pressure [kPa],
$e_a$—actual vapor pressure [kPa],
$e_s - e_a$—saturation vapor pressure deficit [kPa],
$\Delta$—slope vapor pressure curve [kPa/°C],
$\gamma$—psychrometric constant [kPa/°C].

The equation incorporates meteorological data for sunshine hours, air temperature, humidity, and wind speed to calculate $ET_0$. Further details on this calculation can be found in the FAO report [25]. A daily time interval was used.

### 2.3.3. Evaluation of Drought Indices

Usually, the arithmetic mean of all station values is used to represent area value. The same weight is used for each station. In this study, the Thiessen polygon method is applied to calculate the weights of each station. Then, area value was calculated as the weighted average for all stations located in the region. The drought area identified by each drought index was compared with the statistical drought affected area by correlation analysis for the Inner Mongolia Autonomous Region, China. Drought events detected by each drought index were also evaluated by using recorded data of drought disaster events from 15 drought monitoring stations located in the Chinese region of the MP.

### 2.3.4. Drought Analyses

The drought indices, which performed better in the Chinese region of the MP, were applied to detect drought characteristics of the entire MP, such as spatial patterns and temporal characteristics. Spatial and temporal characteristics of drought were detected by empirical orthogonal function (EOF) analysis. One advantage of using the EOF is the ability to identify and quantify the spatial structures of correlated variability [26]. In addition, this method is able to find both temporal and spatial patterns.

2.3.5. Trends Detection for Extreme Cold and Drought

The rank-based non-parametric Mann–Kendall test [27] and trend magnitude method were applied to detect the long-term monotonic trends and their magnitudes. This test can handle non-normality, censoring, data reported as "less-than" values, and seasonality; it also has a high asymptotic efficiency [28]. Further details regarding this test were reported by [29].

## 3. Results and Discussion

### 3.1. Trends of Extreme Cold Indices in the MP

Trends of extreme cold indices in the MP were detected by the Mann–Kendall test for the period of 1969–2017. Overall, it showed a warming climate over the MP during the past 49 years. Extreme air temperature (TXn, TNn, TX_Ann, and TN_Ann) exhibited increasing trends in the study period, whereas the frequency of cold days/nights (TX10p/TN10p), FD, and ID showed decreasing trends. Although changes of climate in the Chinese region were similar to those in the entire MP, those in the Mongolian region differed slightly. In particular, the frequency of TX10p and TN10p increased in the Mongolian region, which indicates increasing numbers of cold days and cold nights in the region. Annual TX and TN increased by 0.30 °C/10 years and 0.42 °C/10 years, respectively (Table 3).

**Table 3.** The magnitude of annual trends for extreme cold indices in the MP during the period 1969–2017.

| Slope | Mongolian Region | Chinese Region | Whole Region | Unit |
|---|---|---|---|---|
| TXn | −0.004 | 0.170 | 0.124 | |
| TNn | 0.018 | 0.375 | 0.280 | °C/10 years |
| TX_Ann | 0.387 | 0.270 | 0.301 | |
| TN_Ann | 0.288 | 0.471 | 0.422 | |
| TX10p | 0.014 | −0.505 | −0.332 | %/10 years |
| TN10p | 0.134 | −1.064 | −0.733 | |
| FD | −1.156 | −3.601 | −2.953 | days/10 years |
| ID | −1.832 | −2.656 | −2.478 | |

The increasing trends of TX and TN in the Mongolian and the Chinese regions were statistically significant at the level of $\alpha = 0.05$ (same as following) in the MP (Figure 2). The significance level of the increasing trend of extreme air temperature in the Chinese region was greater than that in the Mongolian region. On a monthly scale, the coldest day and night exhibited increasing trends for every month in the MP and in its Chinese region. However, decreasing trends were shown for several months in the Mongolian region between March and June (February and August) for the coldest day (night). The annual coldest day (TX_Ann) and coldest night (TN_Ann) showed increasing trends in the MP, both of which were non-significant. The coldest night trends showed a slight decrease in the Mongolian region. Although TX10p and TN10p exhibited decreasing trends, that of TN10p was statistically significant. FD and ID in the MP also showed statistically significant decreasing trends. The trends of these extreme cold indices in the Chinese region were similar to those in the entire MP.

The spatial distribution for trends of extreme cold indices at each station is shown in Figure 3. As previously mentioned, TXn and TNn showed increasing trends in the MP. The figure shows that this increasing trend of TXn was significant at only two stations in the Chinese region and was non-significant at most of these stations. However, that of TNn was significant at half of the 64 stations in the Chinese region. Compared with stations in the Mongolian region, more stations showed increasing trends in the Chinese region. Stations in the Mongolian region with increasing and decreasing trends were approximately equal in number. This result implies that the numbers of annual coldest days and coldest nights mainly increased in the Chinese region but changed little in the Mongolian region. Although the annual coldest days for most of the stations in the Chinese region increased during the past 49 years, all five stations located in the Hulunbuir pasture land showed a decreasing trend; the increasing trend was also significant at one station. Because warming increased

in the MP, the frequency of cold days and cold nights exhibited decreasing trends. These results were similar to the TXn and TNn, such that the decreasing trend of TX10p was non-significant at most of stations in the Chinese region, whereas that of TN10p was significant at 41 of 64 stations in this region. Of the different regions, TX10p and TN10p showed decreasing trends at almost all stations in the Chinese region. For stations in the Mongolian region, the numbers of those with decreasing and increasing trends was also approximately equal. This result means that the frequency of cold days and nights for the Chinese region obviously decreased, whereas that for the Mongolian region changed little. Of 91 stations, 85 exhibited decreasing trends for FD, which was significant at 72 stations. ID also generally showed a decreasing trend at almost all 91 stations except for one station in the north–central region of Mongolia.

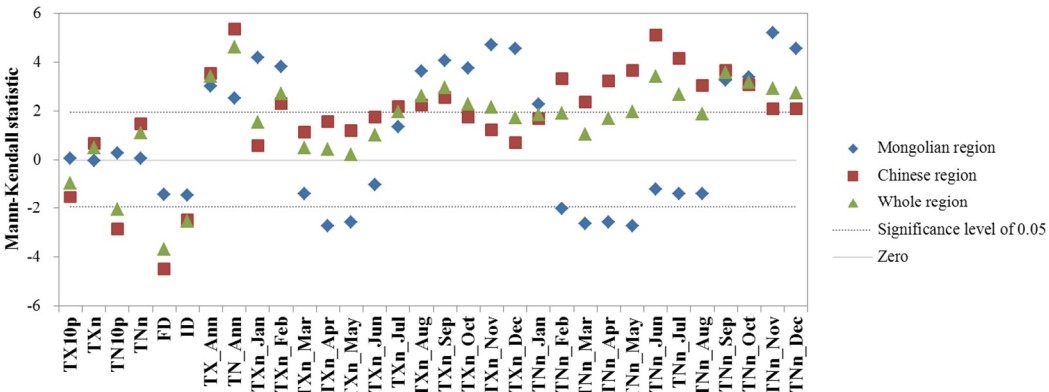

**Figure 2.** Annual and monthly trends of extreme cold indices for different regions during the period 1969–2017.

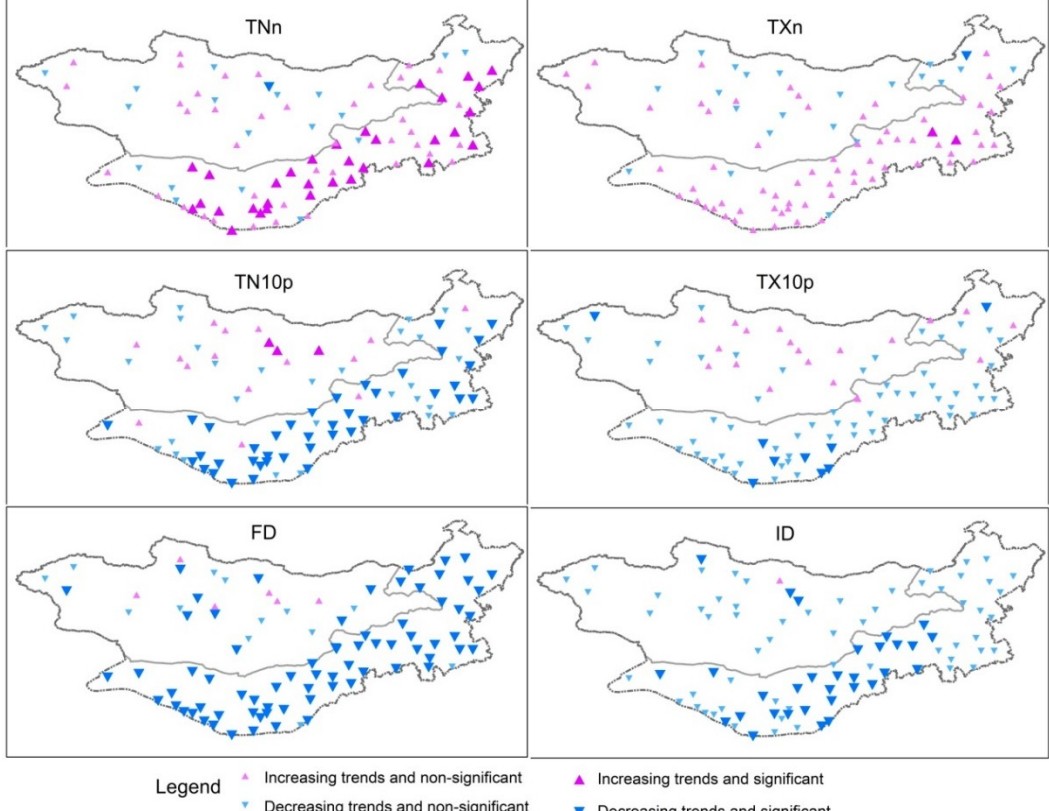

**Figure 3.** Trends of extreme cold indices detected for each station in the MP during the period 1969–2017 (significance level α = 0.05).

## 3.2. Droughts

### 3.2.1. Evaluation of Drought Indices by Using Recorded Historical Drought Data in the Chinese Region

Drought areas identified by each drought index were evaluated by using statistical records of drought areas from 1979 to 2017 in the Inner Mongolia Autonomous Region. The correlation coefficients between annual series of statistical drought areas and those identified by each drought index are shown in Figure 4. Drought areas identified by SPI6 with the criterion of "up to severe drought" showed the strongest correlation with the statistical records; its correlation coefficient was 0.913. SPI6 with the criteria of "up to mild drought" and "moderate drought" also showed the relative stronger correlation than other drought indices when identifying drought areas, with correlation coefficients of 0.836 and 0.747, respectively. In addition, the CMDI with the criterion of "up to mild drought" and SPI3 with that of "up to moderate drought" attained satisfactory results for identifying affected areas of drought, with correlation coefficients of 0.641 and 0.683, respectively. However, these indices showed weak correlation for SPI1 and SPI12 because the correlation coefficients of these drought indices by all criteria were <0.5.

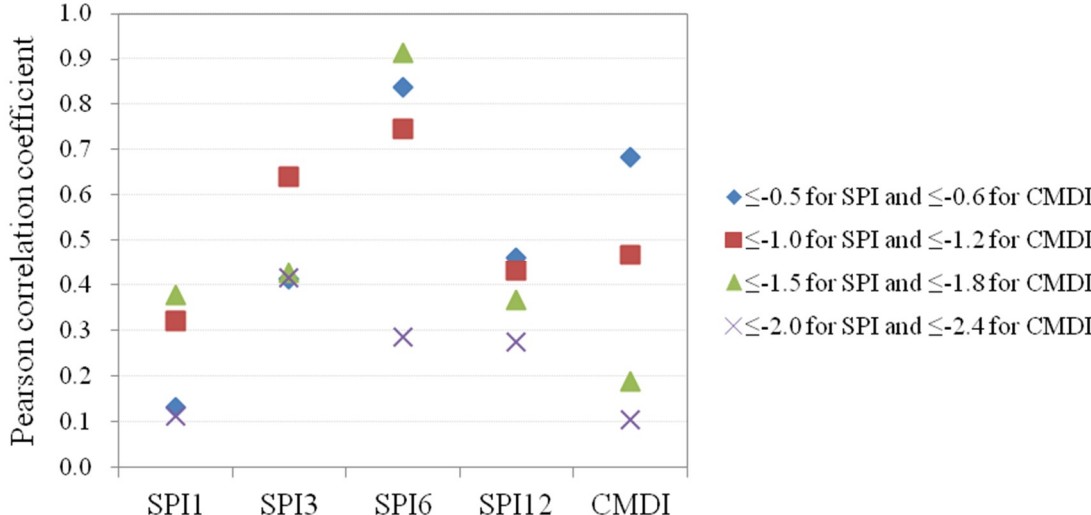

**Figure 4.** Correlation coefficients between annual series of drought area from statistical records and that detected by each drought indices from 1979 to 2017 in the Inner Mongolia province, China.

The numbers of drought events which were detected by drought indices at 15 drought monitoring stations were evaluated by historical drought records from 1991 to 2017 in the Chinese region of the MP (Figure 5). The results show that the five drought indices (SPI1, SPI3, SPI6, SPI12, and CMDI) generally performed well, although some drawbacks associated with each index were noted when used for detecting drought events in the Chinese region of the MP. The SPI12 missed more drought events than other indices. This may have occurred because the SPI12 mainly represents hydrological drought; the historical drought records used in this study were for agrometeorological drought events. For detection of agrometeorological drought events at recording stations, SPI with shorter timescales such as SPI1, SPI3, and SPI6 represented agricultural and meteorological droughts and performed better than the SPI12 with longer time scales. Only slight differences in performance were noted for SPI1, SPI3, and SPI6, which were able to detect more than 60% of the recorded drought events. The CMDI, which is widely applied for monitoring meteorological drought in China, showed the best performance among these drought indices. Of the 228 recorded drought events, 208 were detected by the CMDI. Moreover, this index detected more records than the SPI at every recording station. Overall, the evaluation results showed that the CMDI performed the best when detecting drought events recorded at 15 drought monitoring stations. SPI1, SPI3, and SPI6 also showed relative satisfactory performances for detecting these recorded droughts.

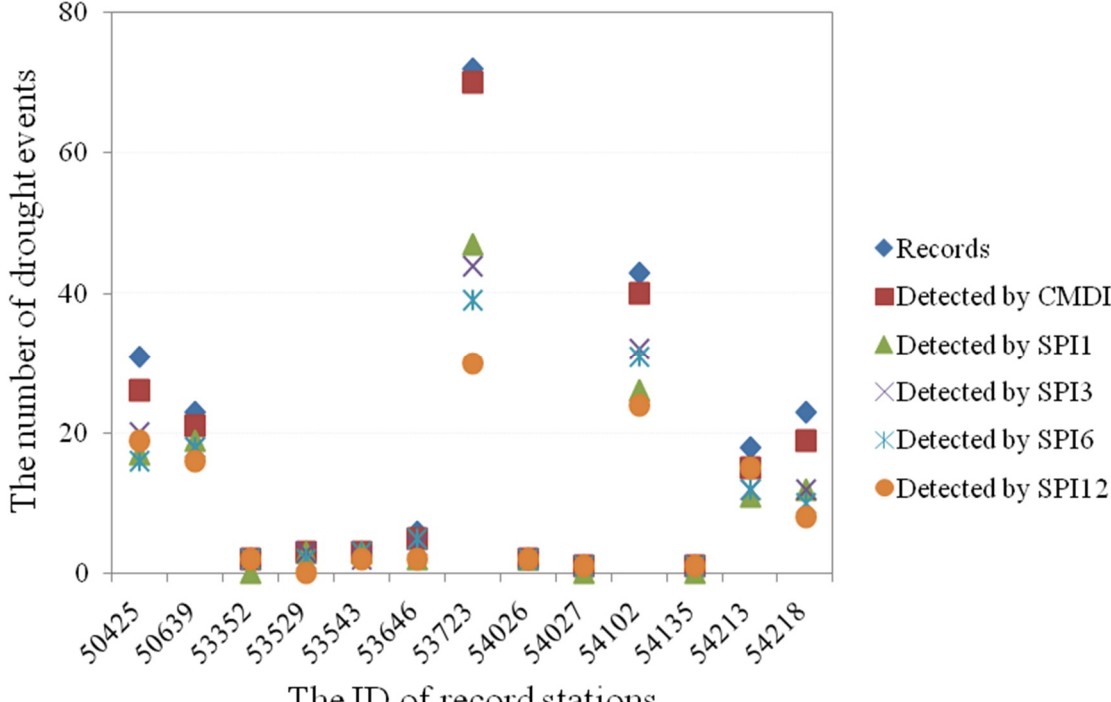

**Figure 5.** Comparison of drought events between records and that detected by each drought indices from 1991 to 2017 in the Chinese region of the MP ("Detected by" means drought events defined by SPI $\leq -0.5$ or CMDI $\leq -0.6$).

Based on the evaluation of drought indices by historical records of drought areas and the numbers of drought events, the CMDI and SPI6 drought indices were selected to analyze drought characteristics of the entire MP because they showed the best performances when detecting drought areas in the Inner Mongolia Autonomous Region and the numbers of drought events in the Chinese region of the MP, respectively.

3.2.2. Detecting Spatial and Temporal Characteristics of Drought by Using the Evaluated Drought Indices (CMDI and SPI6) in the MP

The EOFs were determined by using the SPI6 and CMDI time series from 95 stations. The first and second EOFs (EOF1 and EOF2) of the SPI6 represented approximately 20.1% and 12.0% of the total variance of the original data, respectively, whereas that of the CMDI explained 18.2% and 9.6%, respectively. These two EOFs represented the leading spatial variability of drought. Figure 6 shows spatial patterns of the first and second EOFs for the SPI6 and the CMDI in the MP. In general, high consistency was exhibited between the spatial patterns of the SPI6 and the CMDI. The first EOF displayed a northwest–southeast pattern of drought severity with two center regions. One center region was located near western Xilinhot city, and the other covered Hohhot and Baotou cities, which is the most developed region in the Inner Mongolia Autonomous Region; these regions experienced the highest severity of drought in the MP. The second EOF showed an obvious north–south pattern of drought severity in the MP that was positive in the southern MP and negative in the northern MP; the zero contour was located approximately at the line of 44° N. This EOF also had two severe center regions that were located at the Alashan Plateau and the Tengger Desert, respectively.

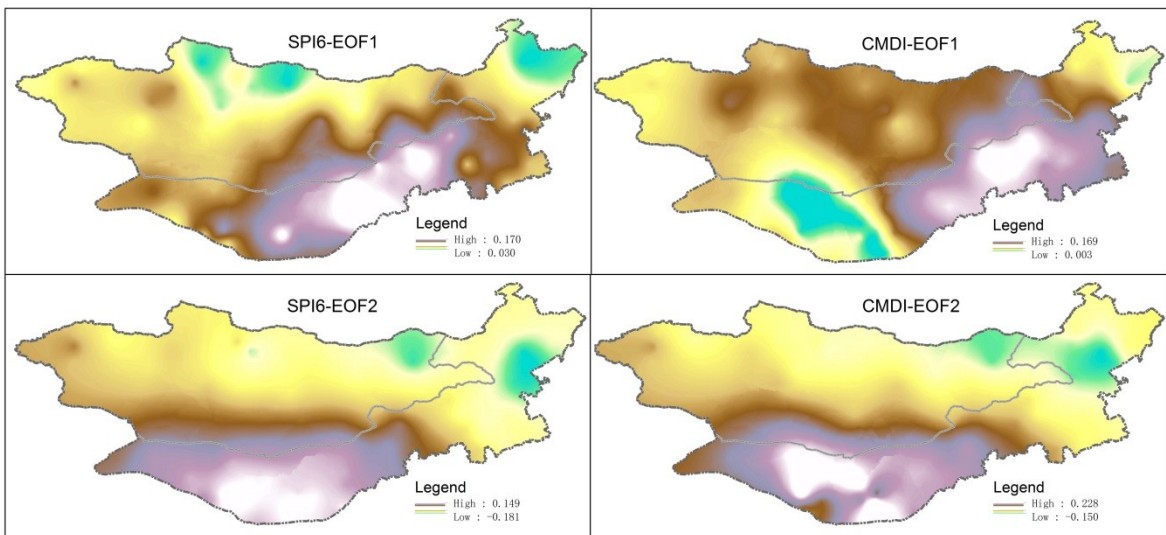

**Figure 6.** Spatial patterns of the first and second EOFs for the SPI6 and the CMDI in the MP.

The numbers of drought years identified by the CMDI, with each criterion from 1969 to 2017 for each month in the MP, are shown in Figure 7. The figure shows the intra-annual variation of drought for 95 stations in the MP. Approximately 15 to 30 drought years were identified in the total 49 years for most stations when using the criterion of "up to mild drought". That is, one-third to two-thirds of the years had mild drought at most regions of the MP during the period 1969–2017. Approximately one-fifth of the years showed "up to moderate drought". As shown in Figure 7, more droughts occurred in the spring months of March, April, and May than other months for both criteria of "up to mild drought" and "up to moderate drought", which indicates that spring droughts were severe in the MP. This result is undesirable for the agriculture of the MP because agricultural water is urgently needed in spring. When applying the criterion of "up to severe drought", droughts in May occurred more than those in other months (Figure 7c); approximately three to five drought years out of the total 49 years in May were noted for most stations. Extreme drought events identified by the CMDI were mainly exhibited in the months of May to August for the MP (Figure 7d). When identified by the SPI6, the number of drought years showed little difference among the months. In general, about 60%, 15%, 6%, and 2% of the years at most stations of the MP showed mild drought, moderate drought, severe drought, and extreme drought, respectively, during the period 1969–2017.

Drought months per year identified by each drought criterion of the SPI6 and the CMDI, from 1969 to 2017 for the 95 stations of the MP, are shown in Figure 8. The criteria of drought included each classification of drought (mild, moderate, severe, and extreme) and the criteria of "up to mild drought", "up to moderate drought", and "up to severe drought". Three to four months per year showed drought at most stations when identified by the SPI6 with the criterion of "up to mild drought", and approximately two drought months per year for the same index with the criterion of "up to moderate drought". When using either of these criteria, the number of drought months identified by the CMDI was obviously larger than that identified by the SPI6. On the contrary, the number of drought months identified by the CMDI was slightly lower than that identified by the SPI6 for the criteria of "up to severe drought" and "up to extreme drought". These results indicate that the CMDI tended to identify more mild droughts and moderate droughts, whereas the SPI6 tended to detect more severe and extreme drought events. This theory was confirmed by the results of drought months per year identified by the criteria of each classification of drought. In general, the MP experienced the most mild drought events and few severe drought or extreme drought events.

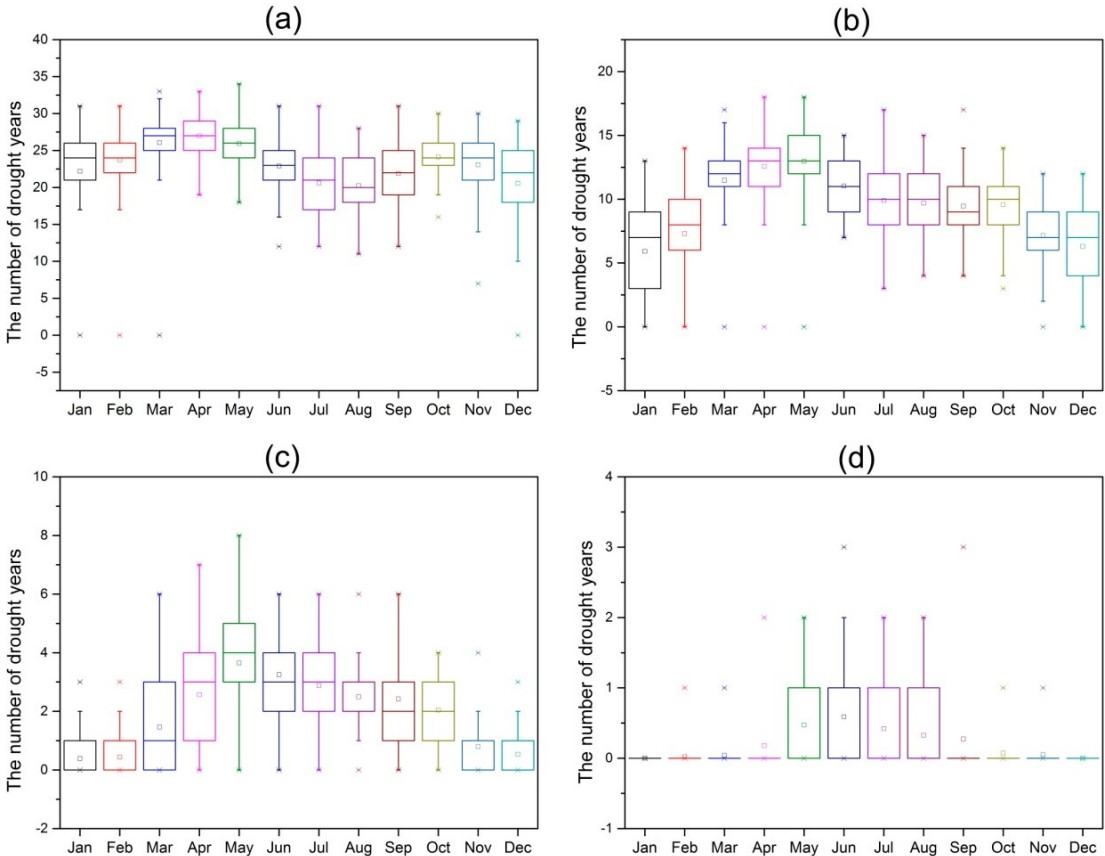

**Figure 7.** The number of drought years identified by the CMDI with the criteria of: (**a**) up to mild drought; (**b**) up to moderate drought; (**c**) up to severe drought; and (**d**) up to extreme drought, from 1969 to 2017 in each month.

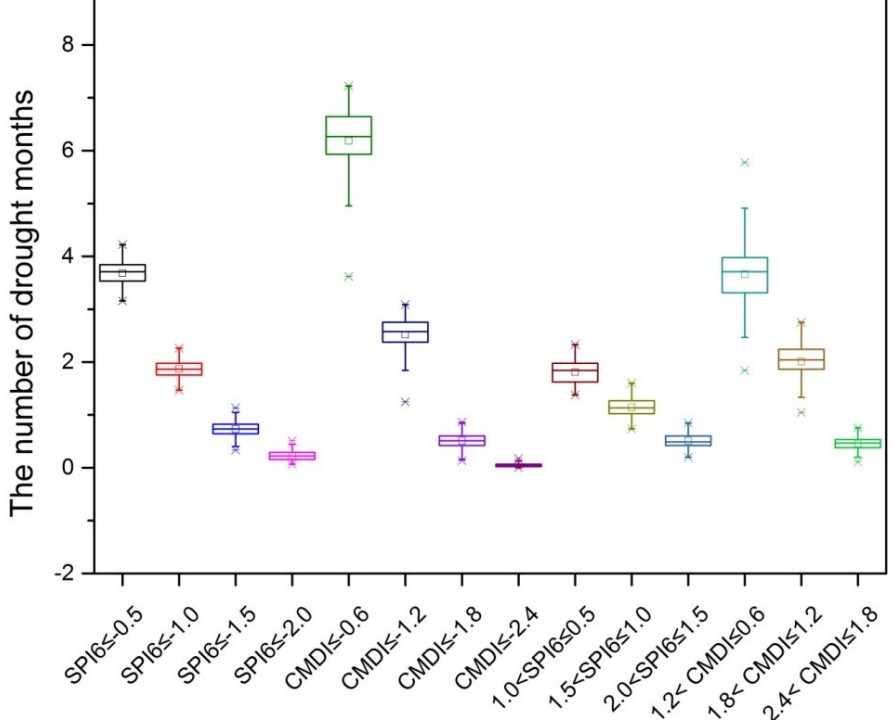

**Figure 8.** Drought months per year identified by each drought criterion of the SPI6 and the CMDI from 1969 to 2017 for 95 stations of the MP.

### 3.2.3. Trends of Meteorological Drought in the MP

The trends of meteorological drought in the MP, represented by the CMDI and the SPI6 for the period of 1969–2017, were detected by using the Mann–Kendall test. The Mann–Kendall statistics for the CMDI, the SPI6, precipitation, and PE are shown in Figure 9. The drought in the MP generally exhibited a decreasing trend for both the CMDI and the SPI6. The trends of drought in the Chinese and Mongolian regions were similar to that in the entire MP. The decreasing trend was statistically significant in the entire MP and the Chinese region for the SPI6. In general, the significance level of the decreasing trend for drought represented by the SPI6 was greater than that expressed by the CMDI. The significance level of the decreasing trend in the Chinese region was greater than that in the Mongolian region, which indicates that drought in the MP was more serious during the past 49 years. Overall, drought in the Chinese region was more enhanced than that in the Mongolian region, and drought expressed by the SPI6 showed greater enhancement than that represented by the CMDI. Annual precipitation showed an increasing trend in the MP, the significance level of which showed only slight difference among regions. It appears that enhanced drought in the MP was not consistent with the increased precipitation for the annual cycle (Figure 9). PE, which represents the ability for water loss, also exhibited an increasing trend in these regions, which is consistent with the enhanced drought in the region. However, the significance level of the increasing trend for PE was lower than that for precipitation.

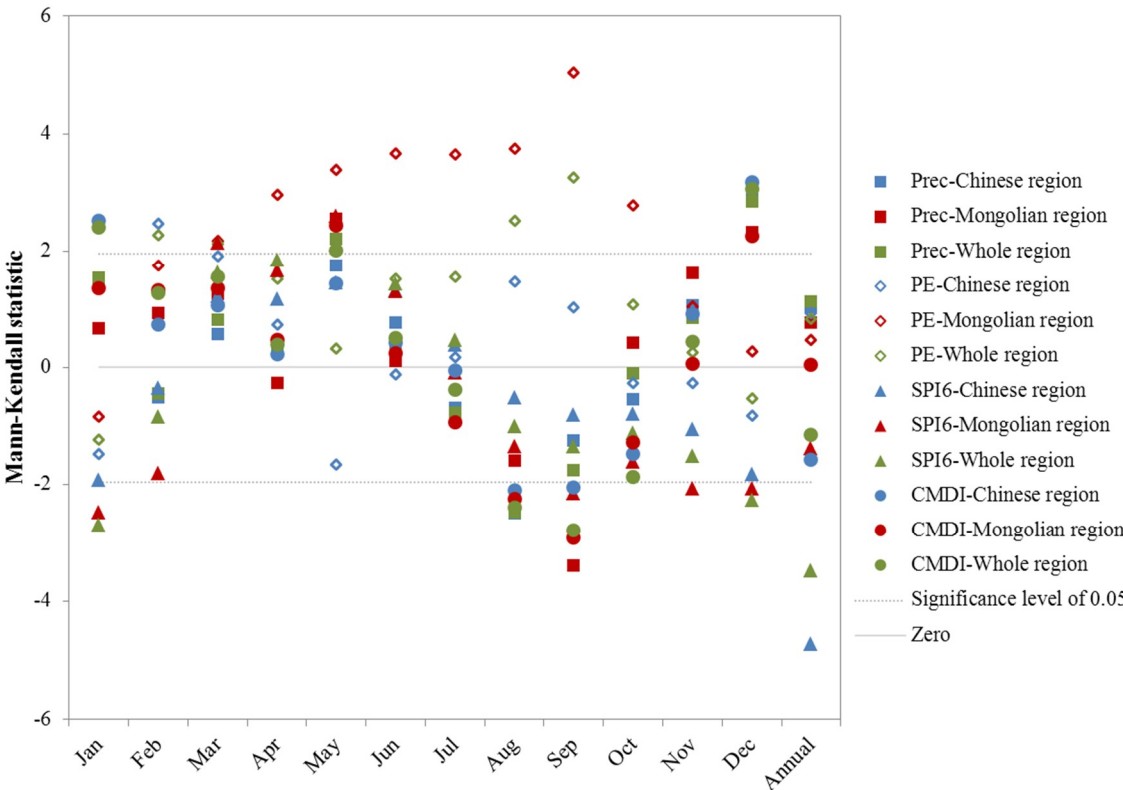

**Figure 9.** Annual and monthly trends of drought, precipitation, and potential evaporation in the MP during the period 1969–2017.

Drought represented by the CMDI and the SPI6 exhibited a decreasing trend from August to October. The significance level of the decreasing trend in the Mongolian region was greater than that in the Chinese region, which was opposite to the annual cycle. The significance level of the decreasing trend for drought represented by the CMDI was greater than that expressed by the SPI6, which was also opposite to the annual cycle. These results indicate that drought was enhanced from August to October, which is consistent with that in the annual cycle. However, this enhanced trend of drought was

greater for the CMDI than that for the SPI6 and was greater in the Mongolian region than that in the Chinese region; the trend was opposite the annual cycle. This result implies that drought during these months was increasingly severe in these regions. Because the water demand in August to October is critical for crops and pastures at some regions of the MP, this drought challenged the agriculture and husbandry in the MP, particularly in the Mongolian region. Although drought was enhanced for the annual cycle, it was weakened from March to June in the MP, including the Chinese and Mongolian regions. As previously mentioned, the spring drought was severe in the MP, which implies that spring drought was also weakened in these regions during the past 49 years. This result was beneficial for agriculture and husbandry in some regions of the MP, where water demand in the spring is critical for crops and pastures. These test results were consistent between the CMDI and the SPI6 for March to June and August to October. The results were opposite between the two indices in January, February, November, and December. In those months, drought represented by the SPI6 showed decreasing trends, whereas that expressed by the CMDI exhibited increasing trends. These results indicate that the SPI6 represents weakened drought, whereas the CMDI tended to express enhanced drought in those months. In general, temperatures were sufficiently cold during those months to inhibit human activities in the MP. Therefore, drought had little effect on agriculture and husbandry during that period.

Overall, monthly precipitation showed similar trends in the two drought indices (Figure 9), which indicates that trends of drought in the monthly cycle were mainly caused by precipitation. Therefore, drought represented by the two indices is consistent with precipitation for the monthly cycle but not for the annual cycle.

The spatial distribution for the Mann–Kendall statistics of the precipitation, PE, the CMDI, and the SPI6 are shown in Figure 10. Overall, the CMDI and SPI6 showed decreasing trends at most stations of the MP, whereas precipitation and PE exhibited increasing trends. The enhanced drought in the Chinese region was more serious than that in the Mongolian region. A decreasing trend was exhibited at most stations of the Chinese region (58 and 53 out of 64 stations for the SPI6 and the CMDI, respectively), whereas a decreasing trend was shown at relatively fewer stations of the Mongolian region (17 and 15 out of 31 stations for the SPI6 and the CMDI, respectively). Drought represented by the SPI6 was more enhanced than that expressed by the CMDI at most stations, which indicates that the SPI6 tended to detect more serious drought than the CMDI in the MP. In the Chinese region, drought represented by the two indices was enhanced more seriously at the Ordos Plateau, the Alashan Plateau, and the Xiliao River basin.

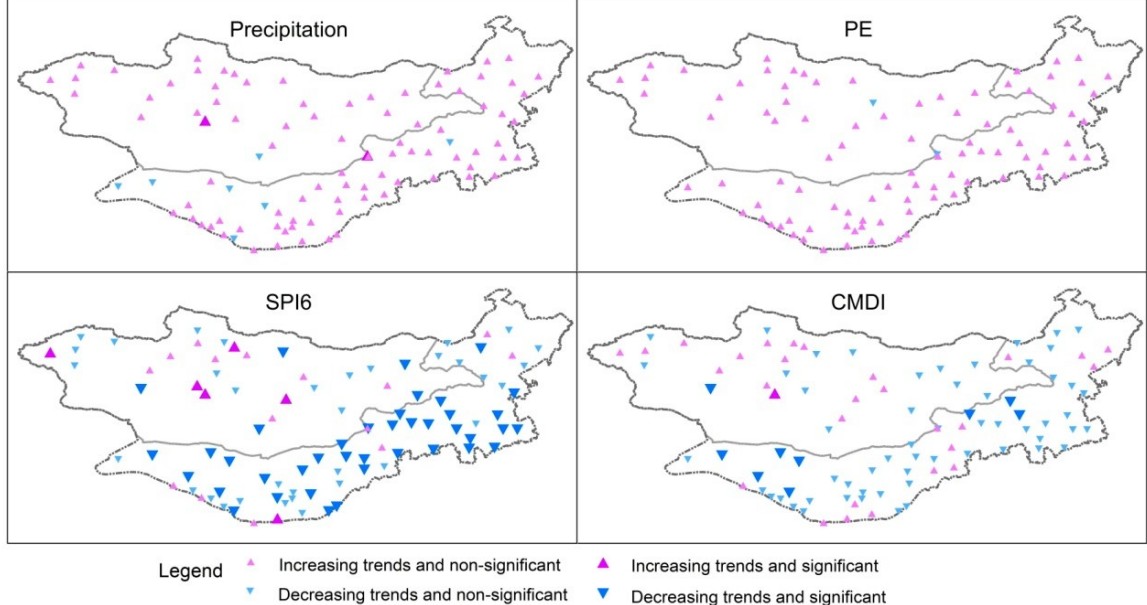

**Figure 10.** Trends of drought, precipitation, and potential evaporation detected for each station during the period 1969–2017 (significance level α = 0.05).

*3.3. Discussion*

Although it showed a warming climate over the MP, the extreme cold events (cold days and cold nights) exhibited an increasing trend. This conclusion was consistent with the results of previous studies for Mongolia [30–32] and the Inner Mongolia Autonomous Region [33]. Extreme cold events are commonly referred to as dzuds in Mongolia. Dzuds are one of main natural disasters in the MP because livestock cannot survive under extreme cold conditions and die in large numbers. Disaster induced by dzuds would be more severe if it is after a summer drought, resulting in inadequate pasture and hay production. Although a warming of the climate has been experienced, the frequency of cold days/nights was increased in Mongolia, and the spring season of Mongolia was becoming colder. These would induce more dzuds and bring more negative effects on herders in Mongolia. The effect of extreme climate change on livestock mortality is a critical aspect for the Mongolian economy.

The drought indexes showed that drought in the MP experienced an increasing trend during the past 49 years. This is consistent with results of [34,35]. The enhanced drought would induce potentially severe consequences for the MP. It could reduce agriculture and pasture production, and then has negative effects on livestock and people's live. Overall, steppe over the MP is facing negative climate change impacts, which induced significant vulnerabilities in the region as a result of extreme weather events, such as droughts and dzuds. Therefore, more research on adaptation strategies for climate change adaptation is needed for Mongolia.

The Siberian High, which is located in Siberia and Mongolia, is the strongest semi-permanent high in the northern hemisphere. It has immense influence on the weather of Asia and Europe, especially for East Asia. Extreme cold over the MP is responsible both for severe winter cold and attendant dry conditions, with little snow across Siberia, Mongolia, and China. The trends of extreme cold indices showed that the climate of the MP has warmed during the past 49 years. This is consistent with the weakening trend of the Siberian High found by [36,37]. These changes would cause immense influences on weather and climate in East Asia. Further research is needed for the mechanism of impacts of these changes on regional weather and climate.

This study used observed station data to represent climate conditions of the MP region. Although nearly a hundred meteorological stations were applied to calculate areal climate variables, the locations of these stations are not perfectly homogeneously distributed in space, especially at the Mongolian region. Inadequate observation network resolution might limit its areal representativeness. Fractal theory can deal with the areal inhomogeneity of a network [38]. It should be noted that monthly precipitation is zero in some months of the stations. The SPI value is treated as zero by the program of the National Drought Mitigation Center. The zero SPI value is treated as the minimum SPI value of a series ($\leq -2.0$) in this study. Loukas and Vasiliades [39] developed two useful procedures in handling null amounts of precipitation. All of these are needed for further research.

## 4. Conclusions

Extreme cold and meteorological drought in the MP were investigated in this study. Several drought indices were evaluated by using recorded historical drought data in the Chinese region of the MP. The evaluated drought indices that performed better at the Chinese region were then applied to detect drought characteristics in the entire MP. The conclusions are presented in this section.

Trends of extreme cold indices showed that the climate of the MP has warmed during the past 49 years. Extreme air temperature (TXn, TNn, TX_Ann, and TN_Ann) increased by 0.12 °C/10 years, 0.28 °C/10 years, 0.30 °C/10 years, and 0.42 °C/10 years, respectively. The frequency of cold days and nights decreased by −0.33%/10 years and −0.73%/10 years, respectively. FD and ID decreased by −2.95 days/10 years and −2.48 days/10 years, respectively. Similar trends of extreme cold indices were exhibited in the Chinese region. In general, the significance level of these trends in the Chinese region was greater than that in the Mongolian region; however, the frequency of cold days/nights was increased in the Mongolian region. The climate of Mongolia showed colder temperatures in the

spring season. Moreover, the coldest day showed a decreasing trend in the Hulunbuir pasture land, which indicates that the coldest day became even colder in this region.

The CMDI and the SPI6 exhibited the best performances when detecting drought areas of the Inner Mongolia Autonomous Region and the number of drought events at 15 stations, respectively. The CMDI tended to identify more mild droughts and moderate droughts, whereas the SPI6 tended to detect more severe and extreme drought events. Spring droughts were more severe than those in other seasons in the MP. Because agricultural water is in urgent demand in spring, this result was undesirable for agriculture in the MP.

In general, drought in the MP was enhanced during the period 1969–2017. Drought in the Chinese region was enhanced greater than that in the Mongolian region. Drought represented by the SPI6 was enhanced more than that expressed by the CMDI. Spring drought showed weakening, which was beneficial for agriculture and husbandry in some regions of the MP where water demand in the spring is critical for crops and pastures. Drought was enhanced from August to October, which is consistent with that in the annual cycle. This enhanced trend of drought was greater for the CMDI than that for the SPI6, greater in the Mongolian region than that in the Chinese region, and opposite to the annual cycle. These results challenged the agriculture and husbandry sectors in the MP, particularly in the Mongolian region.

Drought represented by the two indices was consistent with precipitation for the monthly cycle but not for the annual cycle. In the Chinese region, drought represented by the two indices was enhanced more at the Ordos Plateau, Alashan Plateau, and the Xiliao River basin.

**Author Contributions:** Z.L., Z.Y., and H.H. conceived and designed the research; B.B. and R.W. analyzed the data; Z.L. contributed to writing the manuscript.

**Funding:** This research was funded by the National Key Research and Development Program of China (2016YFC0401306) and National Natural Science Foundation of China (41571027, 41661144030, 41561144012).

**Acknowledgments:** We would like to thank China Meteorological Administration, Climatic Research Unit (https://crudata.uea.ac.uk/), and National Climatic Data Center (ftp://ftp.ncdc.noaa.gov/pub/data/gsod) for the data provided.

**Conflicts of Interest:** The authors declare no conflict of interest.

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
