# Peer review of "Evaluation of Extreme Cold and Drought over the Mongolian Plateau"

_water, doi:10.3390/w11010074_

Round 1

Reviewer 1 Report

The manuscript entitled: Evaluation of extreme cold and drought over the Mongolian Plateau presents important issue of several drought indices were evaluated by analysing recorded historical drought data in the Chinese region of the MP during 1969-2017. Line 139-140, add some information about the Penman–Monteith method, which is recommended by the United Nations Food and Agriculture Organization (FAO). Line 47, it should be explained, what indices You mean, Numerous specialized indices have been proposed to assess the severity of meteorological, hydrological, and agricultural forms of drought [9]. From line 53: The ideal length of precipitation records for SPI 54 calculation is a continuous period of at least 30 years [13]. Because the SPI is a probability-related 55 index, longer record lengths relate to greater confidence in the stability of the underlying statistics. 56 Therefore, it is recommended that the length of precipitation data used in SPI calculation should be 57 as long as possible [14]. So please decide 30 years is ideal time or time as long as possible is ideal?

Line 142, on what base Parameters a, b, and c were taken.  In what programme the Figure 6. Spatial patterns of the first and second EOFs for the SPI6 and the CMDI in the MP. And the rest of the figures, was prepared? Line 418-420. In the following sentence: Trends of extreme cold indices showed that the climate of the MP has warmed during the past 49 years. Extreme air temperature (number median) exhibited increasing trends during the study period, whereas the frequency of cold days/nights, FD (how much?), and ID (number) showed decreasing trends, include some exact results, numbers. Line 438, add values, how much greater? How the results of the present work are expected to help making priorities for drought management?  Some information about practical use of the obtained results, both in the section Results and Conclusions should be underlined. The last point of the article contains in fact only the conclusions relating to the researched case study, but there is no more detailed perspective. Besides, there is no discussion about possible limitations of using the proposed modelling. Therefore, I propose to consider the possibility of completing the last point for discussion on possibilities and limitations of the use of such analysis, but on more broadly sense, in other region, or other parameters.

Author Response

Response to the Reviewers’ Comments

Manuscript ID: water-407925

Article title: Evaluation of extreme cold and drought over the Mongolian Plateau

Author(s): Liu / Yao/ Huang/ Batjav/ Wang

It would be greatly appreciated for your kind reviewing to this paper. The manuscript is thoroughly revised according to your valuable comments and suggestion. The revised manuscript should, after incorporating your valuable advices and suggestions, be improved greatly. For your convenience to re-review the paper, the corrections corresponding to your comments are described in detail as follows:

Comments and Suggestions for Authors

The manuscript entitled: Evaluation of extreme cold and drought over the Mongolian Plateau presents important issue of several drought indices were evaluated by analysing recorded historical drought data in the Chinese region of the MP during 1969-2017.

Line 139-140, add some information about the Penman–Monteith method, which is recommended by the United Nations Food and Agriculture Organization (FAO).

Reply: Thanks for your suggestion. More information about the Penman-Monteith method has been added in this section as following,

The Penman–Monteith method for estimating ETo can be derived as follows:

Where,  ETo - reference crop evapotranspiration [mm/day],

Rn - net radiation at the crop surface [MJ/m2 day],

G - soil heat flux density [MJ/m2 day],

T - mean daily air temperature at 2-m height [°C],

u2 - wind speed at 2-m height [m/s],

es - saturation vapor pressure [kPa],

ea - actual vapor pressure [kPa],

es-ea - saturation vapor pressure deficit [kPa],

Δ - slope vapor pressure curve [kPa/°C],

γ - psychrometric constant [kPa/°C].

       The equation incorporates meteorological data for sunshine hours, air temperature, humidity, and wind speed to calculate ETo. Further details on this calculation can be found in the FAO report [26]. A daily time interval was used.

Line 47, it should be explained, what indices You mean, Numerous specialized indices have been proposed to assess the severity of meteorological, hydrological, and agricultural forms of drought [9].

Reply: Thanks for your comments. We want to express that there are many drought indices have been proposed. These indices can be categorized into three forms, the meteorological, hydrological and agricultural drought indices. These indices can be found as following. However, this study is focus on meteorological drought. Therefore, only meteorological drought indices are described in the manuscript.

Meteorological drought indices: Discrete and cumulative precipitation anomalies, Rainfall deciles, Palmer Drought Severity Index, Drought Area Index, Rainfall Anomaly Index, Standardized Precipitation Index

Hydrological drought indices: Total water deficit, Cumulative streamflow anomaly, Palmer Hydrological Drought Severity Index, Surface Water Supply Index

Agricultural drought indices: Crop Moisture Index, Palmer Moisture Anomaly Index, Computed soil moisture, Soil Moisture Anomaly Index

The sentence “Numerous specialized indices have been proposed to assess the severity of meteorological, hydrological, and agricultural forms of drought [9].” has been revised to “There are many drought indices have been proposed [15]. These indices can be categorized into three forms, the meteorological, hydrological and agricultural drought indices.” to make it more accurate.

From line 53: The ideal length of precipitation records for SPI 54 calculation is a continuous period of at least 30 years [13]. Because the SPI is a probability-related 55 index, longer record lengths relate to greater confidence in the stability of the underlying statistics. 56 Therefore, it is recommended that the length of precipitation data used in SPI calculation should be 57 as long as possible [14]. So please decide 30 years is ideal time or time as long as possible is ideal?

Reply: Thanks for your comments. The ideal length of data series for SPI is as long as possible. We want to represent that the length of data series for SPI is at least 30 years.

The sentence “The ideal length of precipitation records for SPI calculation is a continuous period of at least 30 years” has been revised to “The length of precipitation records for SPI calculation is a continuous period of at least 30 years” to avoid misunderstanding.

Line 142, on what base Parameters a, b, and c were taken.  In what programme the Figure 6. Spatial patterns of the first and second EOFs for the SPI6 and the CMDI in the MP. And the rest of the figures, was prepared?

Reply: Thanks for your comments. This CMDI method is recommended by the National Standard of China (GB/T 20481–2006) for classification of meteorological drought. It is used for drought detection in China by the Public Meteorological Service Center of CMA. The parameters a, b, and c were set to 0.4, 0.4, and 0.8, respectively according to their initial definitions. Therefore, these parameters are constant for the whole China. The same constant values have also been applied in other regions in China, for example, seven large river basins (Qian et al., 2011), the Songnen Plain (Song et al., 2014), the Songhua River Basin (Song et al., 2015),

The sentence “Parameters a, b, and c in Eq. (1) were taken as 0.4, 0.4, and 0.8, respectively” has been revised to “Constants a, b, and c in Eq. (1) were taken as 0.4, 0.4, and 0.8, respectively” to avoid misunderstanding.

The Figure 1, 3, 6, and 10 are made by the ArcGIS 10.2. The Figure 2, 4, 5, and 9 are made by the Microsoft Excel 2010. Figure 7 and 8 are made by the Origin 8.

References:

Qian, W.; Shan, X.; Zhu, Y. Ranking regional drought events in China for 1960-2009. Advances in Atmospheric Sciences, 2011, 28, 310-321.

Song, X.Y.; Li, L.J.; Fu, G.B.; Li, J.Y.; Zhang, A.J.; Liu, W.B.; Zhang, K. Spatial–temporal variations of spring drought based on spring-composite index values for the Songnen Plain, Northeast China. Theor. Appl. Climatol. 2014, 116, 371–384.

Song, X.; Song, S.; Sun, W.; Mu, X.; Wang, S.; Li, J.; Li, Y. Recent changes in extreme precipitation and drought over the Songhua River Basin, China, during 1960–2013. Atmospheric Research, 2015, 157, 137-152.

Line 418-420. In the following sentence: Trends of extreme cold indices showed that the climate of the MP has warmed during the past 49 years. Extreme air temperature (number median) exhibited increasing trends during the study period, whereas the frequency of cold days/nights, FD (how much?), and ID (number) showed decreasing trends, include some exact results, numbers.

Reply: Thanks for your comments. The exact results have been added in this part. The sentence “Extreme air temperature exhibited increasing trends during the study period, whereas the frequency of cold days/nights, FD, and ID showed decreasing trends.” has been revised to “Extreme air temperature (TXn, TNn, TX_Ann, and TN_Ann) increased by 0.12 °C/10a, 0.28 °C/10a, 0.30 °C/10a and 0.42 °C/10a, respectively. The frequency of cold days and nights decreased by -0.33 %/10a and -0.73 %/10a, respectively. FD and ID decreased by -2.95 days/10a and -2.48 days/10a.” to make it more clearly.

Line 438, add values, how much greater? How the results of the present work are expected to help making priorities for drought management?  Some information about practical use of the obtained results, both in the section Results and Conclusions should be underlined. The last point of the article contains in fact only the conclusions relating to the researched case study, but there is no more detailed perspective. Besides, there is no discussion about possible limitations of using the proposed modelling. Therefore, I propose to consider the possibility of completing the last point for discussion on possibilities and limitations of the use of such analysis, but on more broadly sense, in other region, or other parameters.

Reply: Thanks for your comments. The sentence “However, this enhanced trend of drought was greater for the CMDI than that for the SPI6, greater in the Mongolian region than that in the Chinese region, and opposite the annual cycle.” has been revised to “This enhanced trend of drought was greater for the CMDI (Zc=-2.4) than that for the SPI6 (Zc=-1.2), greater in the Mongolian region (Zc=-1.9) than that in the Chinese region (Zc=-1.3), and opposite the annual cycle.” to make it more clearly.

The sentence “These results challenged the agriculture and husbandry sectors in the MP, particularly in the Mongolian region.” in Line 519-520 describes detailed perspective.

A new paragraph is added as the last paragraph of the Discussion section to describe the limitations of the present study due to observation network, data quality and approach. The new paragraph is as following,

“This study used observed station data to represent climate conditions of the MP region. Although nearly a hundred meteorological stations were applied to calculate areal climate variables, the locations of these stations are not perfect homogeneously distributed in space, especially at the Mongolian region. Inadequate observation network resolution might limit its areal representativeness. Fractal theory can deal with the areal inhomogeneity of a network [30], and is needed for further research.”

Reviewer 2 Report

Review of the manuscript entitled “ Evaluation of extreme cold and drought over the Mongolian Plateau   by   Zhaofei Liu , Zhijun Yao , Heqing Huang , Batbuyan Batjav  and Rui Wang

I find the paper a good  contribution to the large and important discussion on  long term climate variations at different regions. The authors  evaluate several  extreme cold and drought indices  to characterize the arid – semiarid Mongolian Plateau (MP) region during the past 49 years (6 different extreme temperature indices: TXn, TNn, TX10p,TN10p, FD, ID  and two drought indices: (SPI, CMDI)

 Extreme air temperature and the frequency of cold days/nights show increasing trends whereas decreasing trends, respetively. The climate of Mongolia  shows colder temperatures in the spring season. Spring droughts are more severe than those in other seasons. Drought represented by the SPI6 was enhanced more than that expressed by the CMDI. Spring drought showed weakening, which was beneficial for agriculture and husbandry in some regions of the MP where water demand in the spring is critical for crops and pastures. Drought was enhanced from August to October and found to be  consistent with  the annual cycle. These results challenged the agriculture and husbandry sectors in the MP, particularly in the Mongolian region.

The paper is well written.

Critical point: The values of the indices were calculated as the weighted average for all stations located in the region. The weights of each station were identified by using the Thiessen polygon method. The drought area identified by each drought index was compared with the statistical drought affected area by correlation analysis for the Inner Mongolia Autonomous Region, China. Drought events detected by each drought index were also evaluated by using recorded data of drought disaster events from drought monitoring stations located in the Chinese region of the MP.

I am skeptical  regarding  the used interpolation  process of the available  network data because  all the physical-mathematical efforts of interpolation to overcome the physiological network weakness are doomed to fail from the start. Inadequate network resolution  could never be recovered by mathematical ingenuity: a basic law of information theory states that ``computers cannot manufacture new information''. So, the only way to get climatic information is to make measurements and to enlarge the available meteorological network. Anyway,  in my opinion this kind of papers is important,because,   beyond the above specific aspects of network weakeness,    offers a first way to   obtain a rapid climatological     information waiting for a denser meteorological network.    For this, the authors are invited to read the paper for     a new next  paper:

Mazzarella A., Tranfaglia G.: Fractal characterisation  of geophysical  measuring  networks  and  its  implications  for  an optimal  location of additive stations: an application to a  rain-gauge network, Theor. Appl. Climatol., 65,157-163, 2000.

Author Response

Response to the Reviewers’ Comments

Manuscript ID: water-407925

Article title: Evaluation of extreme cold and drought over the Mongolian Plateau

Author(s): Liu / Yao/ Huang/ Batjav/ Wang

It would be greatly appreciated for your kind reviewing to this paper. The manuscript is thoroughly revised according to your valuable comments and suggestion. The revised manuscript should, after incorporating your valuable advices and suggestions, be improved greatly. For your convenience to re-review the paper, the corrections corresponding to your comments are described in detail as follows:

Comments and Suggestions for Authors

Review of the manuscript entitled “ Evaluation of extreme cold and drought over the Mongolian Plateau   by   Zhaofei Liu , Zhijun Yao , Heqing Huang , Batbuyan Batjav  and Rui Wang

I find the paper a good  contribution to the large and important discussion on  long term climate variations at different regions. The authors  evaluate several  extreme cold and drought indices  to characterize the arid – semiarid Mongolian Plateau (MP) region during the past 49 years (6 different extreme temperature indices: TXn, TNn, TX10p,TN10p, FD, ID  and two drought indices: (SPI, CMDI)

 Extreme air temperature and the frequency of cold days/nights show increasing trends whereas decreasing trends, respetively. The climate of Mongolia  shows colder temperatures in the spring season. Spring droughts are more severe than those in other seasons. Drought represented by the SPI6 was enhanced more than that expressed by the CMDI. Spring drought showed weakening, which was beneficial for agriculture and husbandry in some regions of the MP where water demand in the spring is critical for crops and pastures. Drought was enhanced from August to October and found to be  consistent with  the annual cycle. These results challenged the agriculture and husbandry sectors in the MP, particularly in the Mongolian region.

The paper is well written.

Critical point: The values of the indices were calculated as the weighted average for all stations located in the region. The weights of each station were identified by using the Thiessen polygon method. The drought area identified by each drought index was compared with the statistical drought affected area by correlation analysis for the Inner Mongolia Autonomous Region, China. Drought events detected by each drought index were also evaluated by using recorded data of drought disaster events from drought monitoring stations located in the Chinese region of the MP.

I am skeptical  regarding  the used interpolation  process of the available  network data because  all the physical-mathematical efforts of interpolation to overcome the physiological network weakness are doomed to fail from the start. Inadequate network resolution  could never be recovered by mathematical ingenuity: a basic law of information theory states that ``computers cannot manufacture new information''. So, the only way to get climatic information is to make measurements and to enlarge the available meteorological network. Anyway,  in my opinion this kind of papers is important,because,   beyond the above specific aspects of network weakeness,    offers a first way to   obtain a rapid climatological     information waiting for a denser meteorological network.    For this, the authors are invited to read the paper for     a new next  paper:

Mazzarella A., Tranfaglia G.: Fractal characterisation  of geophysical  measuring  networks  and  its  implications  for  an optimal  location of additive stations: an application to a  rain-gauge network, Theor. Appl. Climatol., 65,157-163, 2000.

Reply: Thanks very much for your comments and suggestion. We absolutely agree with your opinion. Yes, measurements at point scale are not able to represent actual areal values. As you mentioned that, the only thing we can do is to enlarge the climatic information from the available meteorological network. We agree with “The areal inhomogeneity of a network can be well characterised by its fractal dimension.” as mentioned by Mazzarella and Tranfaglia (2000). We’d like to try this method in our next study.

A new paragraph is added as the last paragraph of the Discussion section to make it more clearly. The new paragraph is as following,

“This study used observed station data to represent climate conditions of the MP region. Although nearly a hundred meteorological stations were applied to calculate areal climate variables, the locations of these stations are not perfect homogeneously distributed in space, especially at the Mongolian region. Inadequate observation network resolution might limit its areal representativeness. Fractal theory can deal with the areal inhomogeneity of a network [39], and is needed for further research.”

In addition, a new reference is also added as following,

Mazzarella, A.; Tranfaglia G. Fractal characterisation of geophysical measuring networks and its implications for an optimal location of additive stations: an application to a rain-gauge network. Theor. Appl. Climatol. 2000, 65, 157163.

Reviewer 3 Report

The introduction needs improvement. It is very short and thus does not provide the necessary background information to the reader.

Data section description needs improvement. Mainly the authors need to elaborate on the meteorological data set quality and the historical drought data spatial resolution. Some sentences need

In section 2.3.2 where drought indices are described, the authors mention that SPI-1 and SPI-3 was calculated. The study region is described as semi-arid with mean annual precipitation of just 246.1mm (lines 33-34). Given this fact, how you apply SPI-1 or SPI-6 given that the probability of no precipitation will not be zero? Have you checked what is the probability of zero precipitation at smaller time scales (1-month, 3-months) for the precipitation time series? This can introduce errors to parameters α and β of Gamma distribution but you do not mention this issue at all in your manuscript. Without describing this issue and how it was addressed it, the methodology about drought indices is not solid.

The discussion section I would incorporate it in the results section forming a new section names as “results and discussion”, of course eliminating repetitions.

The authors do not include any note about the limitations of the present study due to observation network, data quality, methodological approach etc. An analysis is as good as the assumptions, data and methodology used. I would like to see in the discussion section the authors addressing these issues and the implications to their work.

As a general note I would like the authors to improve the content of their manuscript(Introduction, Materials and Methods, and Conclusions). As for the Results and Discussion section I would like to see better presentation and organization. Since their work has the potential to be of higher quality and impact, I recommend major revisions.

More specific comments:

Line 59: It would be good to just mention the data you used in this paragraph since it provides the summary of your effort.

Line 60: “non-parameter” I believe it should be changed to “non-parametric”.

Line 60: please name the statistical test.

Lines 58-65: This paragraph needs improvement. It should be more detailed.

Lines 70-72: Not clear sentence. Needs rephrasing.

Lines 72-73: Winter season is DJF, why do you say "including"? Please rephrase.

Line 75: Not proper use of “amounts”. Please find more appropriate wording.

Line 77: I would like to see the different networks you describe in section 2.2.1 with different symbols at figure 1. The information you provide is not at the level it should be. Please make necessary changes to the map.

Line 84 and line 92: Are these two quality control protocols the same? Please mention the steps of quality control taken by these authorities. Have you personally checked the time series for outliers etc in order to attest for the quality?

Lines 95-97: Please be more specific about this sources of information. What is the spatial level of reporting (e.g. county, municipality etc.), if there is one. Does this information provide an area estimate of the affected area? Are there delineated maps of the affected areas? Not everyone is versed with data sets from these two countries, so you need to elaborate.

Line 97: “The total annual data series was from 1979 to 2017.” It need to change to “The total annual data series record was from 1979 to 2017.”

Line 101: Why do you use only 15 stations? The others do not have drought information?

Lines 103-105: Do you mean both historic data sets? Or just the second one? Please rephrase to be more clear.

Line 139: Expand on the Penman-Monteith method.

Line 143: Are the values recommend for a, b, and c parameters for the study region or there numbers are for the whole China? You need to elaborate in the manuscript in order to support your decision.

Lines 145-148: Did the authors tried other simple spatial interpolation methodologies? I am skeptical of how area value was calculated. Please explain the grounds of why deciding to select this method.

Line 159: Be consistent with how you use nonparametric (elsewhere non-parametric).

Line 160-161: The MK test, indeed can be computed if there are missing values but the performance of the test could be adversely affected. The MK test although is applicable in many situations, it is not without limitations. For example, if the data gaps are great step trend should be used rather than monotonic trend analysis.

Line 288: Please add figure number

Author Response

Response to the Reviewers’ Comments

Manuscript ID: water-407925

Article title: Evaluation of extreme cold and drought over the Mongolian Plateau

Author(s): Liu / Yao/ Huang/ Batjav/ Wang

It would be greatly appreciated for your kind reviewing to this paper. The manuscript is thoroughly revised according to your valuable comments and suggestion. The revised manuscript should, after incorporating your valuable advices and suggestions, be improved greatly. For your convenience to re-review the paper, the corrections corresponding to your comments are described in detail as follows:

Comments and Suggestions for Authors

The introduction needs improvement. It is very short and thus does not provide the necessary background information to the reader.

Reply: Thanks for your comments and suggestion. Several references have been added in the introduction section to give more information to the reader. Added descriptions are as following,

In the first paragraph, (Line 33-37), two sentences “The greatest threat to humans and the natural environment is manifested locally via changes in regional extreme weather and climate events [4,5]. Changes in the frequency or intensity of extreme weather and climate events have profound impacts on both human society and the natural environment [1] because society as a whole is vulnerable to extreme weather and climate [6].” are added.

In the second paragraph, (Line 41-44), two sentences “The consecutive 1999–2002 droughts and dzuds were the worst recorded during the last 50 years and caused 30% of the national herd losses in Mongolia [9]. The 2009–2010 dzud was also very severe in which 8.5 million livestock died in Mongolia, amounting to 20% of the national herd [10](Fernández-Giménez et al., 2012).” are added.

In the third paragraph, (Line 48-51), the sentence “The World Meteorological Organisation Commission for Climatology/Climate Variability held a meeting in Geneva in November 1999 and was the first to recommended 10 simple and feasible indices for climate extremes [12].” is added. (Line 52-55), the sentence “It has been suggested that the worldwide use of accepted climate extreme indices should allow for comparisons with associated information from various regional-scale studies and provide evidence of changes in extreme weather and climate events [14].” is added.

In the fouth paragraph, (Line 59-60), three sentences “The SPI and the PDSI are more popular than other indices [17,18]. Keyantash and Dracup [19] evaluated these meteorological drought indices by applying a weighted set of six evaluation criteria and found that the SPI showed better performance than the PDSI. Hayes et al. [17] found that the SPI improved drought detection and monitoring capabilities over the PDSI when monitoring the 1996 drought in the United States.” are added.

Data section description needs improvement. Mainly the authors need to elaborate on the meteorological data set quality and the historical drought data spatial resolution. Some sentences need

Reply: Thanks for your comments.

In the section “2.2.1 Meteorological observed station data”, (Line 111-114), several new sentences “We evaluate the CRU and the NCDC datasets by the NMO and the NAMEM dataset. Results show that the CRU and the NCDC data are consistent with the NMO and the NAMEM data. Based on data evaluation, we believe that the NCDC and the CRU data are with reliability at the study region.” are added for more descriptions of the meteorological data set quality.

In the section “2.2.2 Recorded historical drought data”, (Line 125-127), several new sentences “The spatial distribution of these stations is shown in Figure 1. These stations are relative well distributed in the Inner Mongolia Autonomous Region. The observation network covers most area of the region.” are added to describe spatial information of the historical drought data.

In section 2.3.2 where drought indices are described, the authors mention that SPI-1 and SPI-3 was calculated. The study region is described as semi-arid with mean annual precipitation of just 246.1mm (lines 33-34). Given this fact, how you apply SPI-1 or SPI-6 given that the probability of no precipitation will not be zero? Have you checked what is the probability of zero precipitation at smaller time scales (1-month, 3-months) for the precipitation time series? This can introduce errors to parameters α and β of Gamma distribution but you do not mention this issue at all in your manuscript. Without describing this issue and how it was addressed it, the methodology about drought indices is not solid.

Reply: Thanks for your comments. Yes, there is zero precipitation in monthly series at the study area. The SPI calculation program is obtained from the National Drought Mitigation Center. In this program, The SPI value is treated as zero for these zero precipitation. This is not correct, because zero precipitation means extreme drought condition, while zero SPI don’t represent extreme drought. We treat these zero SPI values as the minimum SPI value in a calculated series, generally, the minimum SPI value is lower than -2.0, which represents extreme drought condition.

In the first paragraph of the section “2.3.2 Drought indices”, (Line 153-156), several sentences “It should be noted that monthly precipitation is zero in some months of stations. The SPI value is treated as zero by this program. But zero precipitation represents extreme drought condition. Therefore, the zero SPI value is treated as the minimum SPI value of a series (≤−2.0) in this study.” are added to explain this.

The discussion section I would incorporate it in the results section forming a new section names as “results and discussion”, of course eliminating repetitions.

Reply: Thanks for your comments. According to your valuable suggestion, the section “4 Discussion” has been incorporated in the new section “3 Results and Discussion”. Line 173-174, the sentence “This conclusion was consistent with the results of previous studies for Mongolia [22–24] and the Inner Mongolia Autonomous Region [25].” is discussion, and has been moved from the Results section to the new section “3.3 Discussion” (the first paragraph in this section). In addition, a new sentence “Although it showed a warming climate over the MP, the extreme cold events (cold days and cold nights) exhibited an increasing trend.” is also added in the first sentence of this paragraph.

The authors do not include any note about the limitations of the present study due to observation network, data quality, methodological approach etc. An analysis is as good as the assumptions, data and methodology used. I would like to see in the discussion section the authors addressing these issues and the implications to their work.

Reply: Thanks for your comments. A new paragraph is added as the last paragraph of the Discussion section to describe the limitations of the present study due to observation network, data quality and approach. The new paragraph is as following,

“This study used observed station data to represent climate conditions of the MP region. Although nearly a hundred meteorological stations were applied to calculate areal climate variables, the locations of these stations are not perfect homogeneously distributed in space, especially at the Mongolian region. Inadequate observation network resolution might limit its areal representativeness. Fractal theory can deal with the areal inhomogeneity of a network [30], and is needed for further research.”

As a general note I would like the authors to improve the content of their manuscript(Introduction, Materials and Methods, and Conclusions). As for the Results and Discussion section I would like to see better presentation and organization. Since their work has the potential to be of higher quality and impact, I recommend major revisions.

Reply: The manuscript has been thoroughly revised according to your valuable comments and suggestion. Please find details of revisions in point-to-point responds as mentioned above.

As for the new section “3.3 Discussion”, the second and third paragraphs in the original manuscript has been move as the first and second ones in the revised version to make it more clearly. In addition, the last paragraph in the original manuscript is incorporated in the second paragraph of the revised manuscript.

More specific comments:

Line 59: It would be good to just mention the data you used in this paragraph since it provides the summary of your effort.

Reply: Thanks for your comments. The words “on the basis of” has been changed to “by” in this sentence.

Line 60: “non-parameter” I believe it should be changed to “non-parametric”.

Reply: Yes. Thanks for your suggestion. The word “non-parameter” has been changed to “non-parametric” in this sentence.

Line 60: please name the statistical test.

Reply: The name has been added. The words “statistical test” has been changed to “Mann–Kendall test”.

Lines 58-65: This paragraph needs improvement. It should be more detailed.

Reply: Thanks for your comments. More details have been added in this paragraph. The revised paragraph is as following,

“In this study, extreme cold and meteorological drought in the MP are evaluated by meteorological observed data and statistical recorded historical data. Firstly, several extreme temperature indices, defined by ETCCDI, are calculated as extreme cold indices. Trends of extreme cold indices are detected by using a non-parametric Mann–Kendall test method. Their spatial distribution is also investigated. Secondly, drought indices including multiple timescales of the SPI and the comprehensive meteorological drought index (CMDI) are calculated and evaluated by using recorded historical drought data in the Chinese region of the MP. Finally, the evaluated drought indices which performed better in the Chinese region were applied for detecting drought characteristics of the entire MP. Drought characteristics include spatial patterns, temporal characteristics, and trends.”

Lines 70-72: Not clear sentence. Needs rephrasing.

Reply: This sentence “The monthly mean air temperature ranges from −15.2 °C in January to 21.6 °C in July and is lower than 0 °C from November to March of the following year.” has been divided into two sentences “The monthly mean air temperature ranges from −15.2 °C in January to 21.6 °C in July. It is lower than 0 °C from November to March of the following year.” to make it more clear.

Lines 72-73: Winter season is DJF, why do you say "including"? Please rephrase.

Reply: Thanks for your comments. The words “including December, January, and February” has been replaced by “(DJF)”.

Line 75: Not proper use of “amounts”. Please find more appropriate wording.

Reply: The words “amounts to” have been replaced by “takes up”.

Line 77: I would like to see the different networks you describe in section 2.2.1 with different symbols at figure 1. The information you provide is not at the level it should be. Please make necessary changes to the map.

Reply: The different networks described in section 2.2.1 have been represented by different symbols in Figure 1. It includes five data networks, drought monitor station, NAMEM station, NCDC station, CRU station, and NMO station. The new Figure 1 is as following, or can be found in the revised manuscript.

Figure 1. Location of the Mongolian Plateau, meteorological stations and drought monitor stations

Line 84 and line 92: Are these two quality control protocols the same? Please mention the steps of quality control taken by these authorities. Have you personally checked the time series for outliers etc in order to attest for the quality?

Reply: The instructions of the NMO dataset describe that “the dataset has passed its data quality control”. But it doesn’t give more details about what quality control processes and how to do this quality control. This dataset is controlled by China Meteorological Administration, and has been widely used for climate change studies in China.

The second dataset (Line 92) is obtained from the National Agency for Meteorology and Environment Monitoring (NAMEM), Mongolia. This dataset only includes annual precipitation data series. We have evaluated it by an outlier detection method. It calculates the mean (μ) and the standard deviation (σ) of the data series. Data that are outside the limits of μ ± 3σ are deemed to be outliers. Evaluation results showed that there is none outlier in this annual series. Line 91-92, the sentence “The dataset is passed quality control.” is deleted to avoid misunderstanding.

Lines 95-97: Please be more specific about this sources of information. What is the spatial level of reporting (e.g. county, municipality etc.), if there is one. Does this information provide an area estimate of the affected area? Are there delineated maps of the affected areas? Not everyone is versed with data sets from these two countries, so you need to elaborate.

Reply: This dataset is Statistical Yearbook Data obtained from the National Bureau of Statistics of China. The spatial level of this statistical data is province and municipality. It includes the drought affected area, but don’t have any delineated maps of the affected areas. It is only statistical data for each province (municipality).

A new sentence “It includes statistics of the drought affected area for each province (municipality).” is added to make it more clearly.

The word “Natural” in Line 97 is wrong. It should be “National”. Therefore, the words “Natural Bureau of Statistics of China” is revised to “National Bureau of Statistics of China”.

Line 97: “The total annual data series was from 1979 to 2017.” It need to change to “The total annual data series record was from 1979 to 2017.”

Reply: Thanks for your comments. The words “The total annual data series was from 1979 to 2017” has been replaced by “The total annual data series record was from 1979 to 2017”.

Line 101: Why do you use only 15 stations? The others do not have drought information?

Reply: Yes. Although there are many meteorological observed stations, there are only 15 drought monitoring stations in the study region. Other meteorological stations don’t have drought information. Therefore, we used these 15 drought monitoring stations record data to evaluate drought indices.

Lines 103-105: Do you mean both historic data sets? Or just the second one? Please rephrase to be more clear.

Reply: It is just for the second one. The drought affect area is also included in the second dataset. But the affect area only has qualitative descriptions, for example, “the affect area is more than 1000,000 ha.” or “the affect area is more than 100,000 ha.”.

The sentence “…the number of drought events was applied in this study.” is changed to “…. the number of drought events was applied as this second drought dataset.” to make it more clearly.

Line 139: Expand on the Penman-Monteith method.

Reply: More information about the Penman-Monteith method has been added in this section as following,

The Penman–Monteith method for estimating ETo can be derived as follows:

Where,  ETo - reference crop evapotranspiration [mm/day],

Rn - net radiation at the crop surface [MJ/m2 day],

G - soil heat flux density [MJ/m2 day],

T - mean daily air temperature at 2-m height [°C],

u2 - wind speed at 2-m height [m/s],

es - saturation vapor pressure [kPa],

ea - actual vapor pressure [kPa],

es-ea - saturation vapor pressure deficit [kPa],

Δ - slope vapor pressure curve [kPa/°C],

γ - psychrometric constant [kPa/°C].

       The equation incorporates meteorological data for sunshine hours, air temperature, humidity, and wind speed to calculate ETo. Further details on this calculation can be found in the FAO report [26]. A daily time interval was used.

Line 143: Are the values recommend for a, b, and c parameters for the study region or there numbers are for the whole China? You need to elaborate in the manuscript in order to support your decision.

Reply: This CMDI method is recommended by the National Standard of China (GB/T 20481–2006) for classification of meteorological drought. It is used for drought detection in China by the Public Meteorological Service Center of CMA. The parameters a, b, and c were set to 0.4, 0.4, and 0.8, respectively according to their initial definitions. Therefore, these parameters are constant for the whole China. The same constant values have also been applied in other regions in China, for example, seven large river basins (Qian et al., 2011), the Songnen Plain (Song et al., 2014), the Songhua River Basin (Song et al., 2015),

The sentence “Parameters a, b, and c in Eq. (1) were taken as 0.4, 0.4, and 0.8, respectively” has been revised to “Constants a, b, and c in Eq. (1) were taken as 0.4, 0.4, and 0.8, respectively” to avoid misunderstanding.

References:

Qian, W.; Shan, X.; Zhu, Y. Ranking regional drought events in China for 1960-2009. Advances in Atmospheric Sciences, 2011, 28, 310-321.

Song, X.Y.; Li, L.J.; Fu, G.B.; Li, J.Y.; Zhang, A.J.; Liu, W.B.; Zhang, K. Spatial–temporal variations of spring drought based on spring-composite index values for the Songnen Plain, Northeast China. Theor. Appl. Climatol. 2014, 116, 371–384.

Song, X.; Song, S.; Sun, W.; Mu, X.; Wang, S.; Li, J.; Li, Y. Recent changes in extreme precipitation and drought over the Songhua River Basin, China, during 1960–2013. Atmospheric Research, 2015, 157, 137-152.

Lines 145-148: Did the authors tried other simple spatial interpolation methodologies? I am skeptical of how area value was calculated. Please explain the grounds of why deciding to select this method.

Reply: This part describes how area value was calculated from station values. Usually, the arithmetic mean of all station values is used to represent area value. The same weight is used for each station. In this study, the Thiessen polygon method is applied to calculate the weights of each station. The weight is different for each station.

The sentences “Area value was calculated as the weighted average for all stations located in the region. The weights of each station were identified by using the Thiessen polygon method.” is revised to “Usually, the arithmetic mean of all station values is used to represent area value. The same weight is used for each station. In this study, the Thiessen polygon method is applied to calculate the weights of each station. Then, area value was calculated as the weighted average for all stations located in the region.” to make it more clearly.

Line 159: Be consistent with how you use nonparametric (elsewhere non-parametric).

Reply: The word “nonparametric” has been changed to “non-parametric”.

Line 160-161: The MK test, indeed can be computed if there are missing values but the performance of the test could be adversely affected. The MK test although is applicable in many situations, it is not without limitations. For example, if the data gaps are great step trend should be used rather than monotonic trend analysis.

Reply: The words “missing values,” has been deleted.

Line 288: Please add figure number

Reply: The figure number has been added. The words “the figure” is changed to “Figure 7”.

Round 2

Reviewer 3 Report

The authors have improved the manuscript a lot and addressed all my comments adequately but one. I would like the authors to address the problem of zero precipitation in the monthly series since the SPI-1 calculation will have errors. My previous comment is the following:

 "In section 2.3.2 where drought indices are described, the authors mention that SPI-1 and SPI-3 was calculated. The study region is described as semi-arid with mean annual precipitation of just 246.1mm (lines 33-34). Given this fact, how you apply SPI-1 or SPI-6 given that the probability of no precipitation will not be zero? Have you checked what is the probability of zero precipitation at smaller time scales (1-month, 3-months) for the precipitation time series? This can introduce errors to parameters α and β of Gamma distribution but you do not mention this issue at all in your manuscript. Without describing this issue and how it was addressed it, the methodology about drought indices is not solid."

Authors Reply: Thanks for your comments. Yes, there is zero precipitation in monthly series at the study area. The SPI calculation program is obtained from the National Drought Mitigation Center. In this program, The SPI value is treated as zero for these zero precipitation. This is not correct, because zero precipitation means extreme drought condition, while zero SPI don’t represent extreme drought. We treat these zero SPI values as the minimum SPI value in a calculated series, generally, the minimum SPI value is lower than -2.0, which represents extreme drought condition. In the first paragraph of the section “2.3.2 Drought indices”, (Line 151-154), several sentences “It should be noted that monthly precipitation is zero in some months of stations. The SPI value is treated as zero by this program. But zero precipitation represents extreme drought condition. Therefore, the zero SPI value is treated as the minimum SPI value of a series (≤−2.0) in this study.” are added to explain this.

My comment on your reply:

The SPI program from  the National Drought Mitigation Center is treating missing daily values as zeros. Text from the manual: "Daily data are summed by month (either calendar month or 4-week period, depending on which aggregate type [monthly or weekly] is being used). Thus, missing daily data values, when being aggregated into monthly sums, are effectively treated as zeroes. For example, if 29 out of 31 input daily values for January 2003 were -99s, the value that would be converted to a 1-month SPI for January 2003 would be the sum of the two good values from that month. If an entire time step period (either a week or month) is filled with missing data values, any resulting output that has that particular time step in its aggregation period will be a missing data value." You state that zero precipitation (during a month) denotes drought, but this is not necessarily correct. 

Please read the following article and more specifically section 3.2 that discusses the issue of zero occurrences in monthly precipitation.

A. Loukas, L. Vasiliades. Probabilistic analysis of drought spatiotemporal characteristics inThessaly region, Greece. Natural Hazards and Earth System Science, Copernicus Publications on behalf of the European Geosciences Union, 2004, 4 (5/6), pp.719-731.

Author Response

Response to the Reviewers’ Comments

Manuscript ID: water-407925

Article title: Evaluation of extreme cold and drought over the Mongolian Plateau

Author(s): Liu / Yao/ Huang/ Batjav/ Wang

Comments and Suggestions for Authors

The authors have improved the manuscript a lot and addressed all my comments adequately but one. I would like the authors to address the problem of zero precipitation in the monthly series since the SPI-1 calculation will have errors. My previous comment is the following:

 "In section 2.3.2 where drought indices are described, the authors mention that SPI-1 and SPI-3 was calculated. The study region is described as semi-arid with mean annual precipitation of just 246.1mm (lines 33-34). Given this fact, how you apply SPI-1 or SPI-6 given that the probability of no precipitation will not be zero? Have you checked what is the probability of zero precipitation at smaller time scales (1-month, 3-months) for the precipitation time series? This can introduce errors to parameters α and β of Gamma distribution but you do not mention this issue at all in your manuscript. Without describing this issue and how it was addressed it, the methodology about drought indices is not solid."

Authors Reply: Thanks for your comments. Yes, there is zero precipitation in monthly series at the study area. The SPI calculation program is obtained from the National Drought Mitigation Center. In this program, The SPI value is treated as zero for these zero precipitation. This is not correct, because zero precipitation means extreme drought condition, while zero SPI don’t represent extreme drought. We treat these zero SPI values as the minimum SPI value in a calculated series, generally, the minimum SPI value is lower than -2.0, which represents extreme drought condition. In the first paragraph of the section “2.3.2 Drought indices”, (Line 151-154), several sentences “It should be noted that monthly precipitation is zero in some months of stations. The SPI value is treated as zero by this program. But zero precipitation represents extreme drought condition. Therefore, the zero SPI value is treated as the minimum SPI value of a series (≤−2.0) in this study.” are added to explain this.

My comment on your reply:

The SPI program from  the National Drought Mitigation Center is treating missing daily values as zeros. Text from the manual: "Daily data are summed by month (either calendar month or 4-week period, depending on which aggregate type [monthly or weekly] is being used). Thus, missing daily data values, when being aggregated into monthly sums, are effectively treated as zeroes. For example, if 29 out of 31 input daily values for January 2003 were -99s, the value that would be converted to a 1-month SPI for January 2003 would be the sum of the two good values from that month. If an entire time step period (either a week or month) is filled with missing data values, any resulting output that has that particular time step in its aggregation period will be a missing data value." You state that zero precipitation (during a month) denotes drought, but this is not necessarily correct. 

Please read the following article and more specifically section 3.2 that discusses the issue of zero occurrences in monthly precipitation.

A. Loukas, L. Vasiliades. Probabilistic analysis of drought spatiotemporal characteristics inThessaly region, Greece. Natural Hazards and Earth System Science, Copernicus Publications on behalf of the European Geosciences Union, 2004, 4 (5/6), pp.719-731.

Authors Reply: Thanks for your comments. Yes, as you and the manual mentioned that, “the zero precipitation denotes drought” is not absolutely correct. But we have checked that all zero monthly precipitation values were in dry seasons. In other words, these zero values are more likely actual conditions than be missing data value. In addition, the first method proposed by Loukas and Vasiliades (2004) treat the null precipitation with a small amount of precipitation, for example 1mm. This is similar with “the zero precipitation denotes drought”.

Overall, in the first paragraph of the section “2.3.2 Drought indices”, (Line 151-154), the sentence “But zero precipitation represents extreme drought condition.” has been deleted to avoid misunderstanding. The sentences “It should be noted that monthly precipitation is zero in some months of stations. The SPI value is treated as zero by this program. Therefore, the zero SPI value is treated as the minimum SPI value of a series (−2.0) in this study.” are moved to the last paragraph of the section 3.3. Discussion. In addition, the sentences “Loukas and Vasiliades [40] developed two useful procedures in handling null amounts of precipitation. All of these are needed for further research.” are added.

Added reference,

Loukas, A.; Vasiliades, L. Probabilistic analysis of drought spatiotemporal characteristics in Thessaly region, Greece. Nat. Hazards Earth Syst. Sci. 2004, 4, 719–731.